# Investigation of Cation Exchange Behaviors of FA_x_MA_1−x_PbI_3_ Films Using Dynamic Spin-Coating

**DOI:** 10.3390/ma14216422

**Published:** 2021-10-26

**Authors:** Hyang Mi Yu, Byeong Geun Jeong, Dae Young Park, Seong Chu Lim, Gon Namkoong, Mun Seok Jeong

**Affiliations:** 1Department of Energy Science, Sungkyunkwan University, Suwon 16419, Korea; gidal0072@skku.edu (H.M.Y.); zinzza228@skku.edu (B.G.J.); 2Department of Physics, Hanyang University, Seoul 04763, Korea; parkdy004@hanyang.ac.kr; 3Department of Smart Fab. Technology, Sungkyunkwan University, Suwon 16419, Korea; 4Applied Research Centre, Department of Electrical and Computer Engineering, Old Dominion University, 12050 Jefferson Avenue, Newport News, VA 23693, USA; 5Department of Energy Engineering, Hanyang University, Seoul 04763, Korea

**Keywords:** cation exchange, dynamic spin-coating, FA_x_MA_1__−__x_PbI_3_, δ-phase FAPbI_3_, FA cations, MA cations

## Abstract

In this study, we fabricated and characterized uniform multi-cation perovskite FA_x_MA_1−x_PbI_3_ films. We used the dynamic spin-coating method to control the cation ratio of the film by gradually increasing the FA+, which replaced the MA+ in the films. When the FA+ concentration was lower than xFA ~0.415 in the films, the stability of the multi-cation perovskite improved. Above this concentration, the film exhibited δ-phase FAPbI_3_ in the FA_x_MA_1−x_PbI_3_ films. The formation of δ-phase FAPbI_3_ disturbed the homogeneity of the photoluminescence spatial distribution and suppressed the absorption spectral bandwidth with the increasing bandgap. The precise control of the cation ratio of multi-cation perovskite films is necessary to optimize the energy-harvesting performance.

## 1. Introduction

Recent studies have shown that mono-cation methylammonium lead triiodide (CH_3_NH_3_PbI_3_, MAPbI_3_) perovskite has some disadvantages, such as narrow absorption spectral bandwidths and weak long-term stability in the application of energy-harvesting devices [1,2,3]. A significant research effort has been made to overcome these problems by synthesizing the multi-cation materials [4,5]. For instance, blending formamidinium (FA) lead triiodide (HC(NH2)_2_PbI_3_, FAPbI_3_) with MAPbI_3_ partially replaces the MA with larger FA cations; thus, both the MA and the FA cations are present. The optical and structural properties are subject to changes, depending on the relative ratio of the multiple cations. For example, when the mixing of MAPbI_3_ (1.64 eV) and FAPbI_3_ (1.43 eV) is controlled, the bandgap is tuned to ~1.48 eV, which is within the near-infrared range [6,7]. However, the multi-cation materials require more attention to optimize their properties. At a particular mixing of the cation ratio, FAPbI_3_ develops into two different phases: the photoactive black perovskite phase (α-FAPbI_3_) and a yellow non-perovskite hexagonal phase (δ-FAPbI_3_) [5,6]. However, the former can easily transform into the latter at room temperature with a significantly broad bandgap of 2.5 eV [5,6]. Therefore, formulating a multi- cation perovskite system using FA and MA cations requires a balanced ratio to obtain a multi-cation FA_x_MA_1__−__x_PbI_3_ perovskite film with enhanced solar-cell efficiency; this has not been fully understood yet.

Among the several synthesis methods, static or dynamic spin-coating has been widely adopted as a facile way to obtain high-quality perovskite films [8,9]. In particular, dynamic spin-coating provides better perovskite film than the static method, which was widely used in the developing stage [9,10]. The static process involves coating the substrate with a perovskite precursor before spinning occurs, resulting in rapid evaporation of the perovskite solvents, anisotropic growth, fast crystallization, and nonuniform perovskite layers [9,10]. Conversely, dynamic spin-coating uses various perovskite precursors that continuously drop onto a rotating substrate [10]. Consequently, solvent evaporation is limited, and the intermediate phase is prolonged, resulting in better mixing of the perovskite elements. Additionally, dynamic spin-coating allows for (i) the solvent incompatibility of different precursors to be resolved; (ii) the separate optimization for each precursor and deposition step; and (iii) different deposition methods to be used for each deposition step. 

Dynamic spin-coating provides a more efficient method for synthesizing uniform perovskite films, but the understanding of the synthesis mechanism is lacking when the dynamic spin-coating is applied to multi-cation perovskite systems. 

In our study, we fabricated a multi-cation FA_x_MA_1-x_PbI_3_ film at different concentrations of FA cations. We used the dynamic spin-coating method because it enabled us to control the amount and the type of organic cations effectively. By increasing the FA concentration, we characterized the multi-cation perovskite films using photoluminescence (PL) and X-ray diffraction (XRD). At an FA concentration xFA of ~0.415, the films showed a narrow bandgap and luminescence peak in PL and a narrow, full-width, half-maximum during XRD. Above this concentration, the film performance degraded with a wide bandgap of 2.5 eV, owing to the formation of δ-phase FAPbI_3_.

## 2. Materials and Methods

### 2.1. Material and Sample Preparation

#### 2.1.1. Preparation of Perovskite Precursor Solutions

First, the mixed perovskite precursor solution was prepared by dissolving 25 mg of FAI (Greatcell Solar (Queanbeyan, Australia)) and 12.5 mg of FAI (Greatcell Solar) in a 1 mL isopropyl alcohol (IPA) solution with 20 mg of MAI (Greatcell Solar (Queanbeyan, Australia)), respectively. In addition, 150 mg of PbI_2_ (lead iodide, 99.9985%) in the mixed solvents of 220 μL dimethylformamide (DMF, Sigma-Aldrich (Seoul, Korea), anhydrous, 99.8%) and 20 μL dimethylsulfoxide (DMSO, Sigma Aldrich (Seoul, Korea)) was also prepared. These two solutions were prepared in a glove box (glove box condition, O_2_: 0% and H_2_0: 0%). Afterwards, both solutions were applied on a hot plate at 70 °C for 24 h using magnetic stirring. 

#### 2.1.2. Fabrication of FA_x_MA_1__−__x_PbI_3_ Film

The FAMAPbI_3_ perovskite films were fabricated by a two-step, dynamic spin-coating procedure on the prepared FTO on glass substrates. The FA_x_MA_1__−__x_PbI_3_ perovskite film was fabricated as follows. The FTO on the glass substrate was cleaned sequentially in an ultrasonic bath for 10 min each in acetone, DI water, and IPA, respectively. Afterwards, the PbI_2_ perovskite precursor solution was coated on the FTO/glass substrate by using the dynamic spin-coating method at 3000 rpm for 30 s. As a result, the transparent yellow color of the PbI_2_ thin film was formed. Then, the PbI_2_ film was heated at 120 °C for 5 min on a hot plate. As a result, the yellow color of the PbI_2_ film was obtained, as shown in the photo image in Figure 1. Finally, the mixed FAI–MAI solution of 50 μL was continuously dropped onto the pre-coated PbI_2_ films at 8 s intervals during spin-coating at 4000 rpm for 30 s, then annealed at 150 °C for 10 min in an air environment. With increasing spin-coating cycles, only FA_x_MA_1__−__x_PbI_3_ films with 25 mg/mL showed a gradual color change from black to bright red, whereas the films with the 12.5 mg/mL FAI remained a black color (Figure 1).

### 2.2. Characterization 

Field-emission scanning electron microscope (SEM) (JSM7000F) measurement was performed to investigate the surface morphologies of the FA_x_MA_1__−__x_PbI_3_ films. The SEM uses a Schottky-type field-emission gun for the electron source and state-of-the-art computer technology for the image-display system. The X-ray diffraction measurements were performed using an X-ray diffractometer (Rigaku, SmartLab) with Cu-Ka radiation (λ = 1.54059 Å). The samples were scanned from 2.5° to 20° at a scan rate of 4°/min with a step size of 0.02°. The optical properties were investigated using an optical microscope system (NT-MDT) with the high-magnification objective lens (NA = 0.7). The excitation laser wavelength of 405 nm and the gratings with 150 grooves were used for the PL spectra of the perovskite films. Moreover, an ultraviolet-visible (UV–Vis) absorption spectrometer [JASCO V-670] was used to measure the chemical properties of the FA_x_MA_1__−__x_PbI_3_ films. 

## 3. Results

Figure 1a describes a schematic diagram of the two-step, dynamic spin-coating method of FA_x_MA_1__−__x_PbI_3_ film using the mixed FAI–MAI precursor solution. The dynamic spin-coating technology was used for continuous dispensing of the precursor solutions rather than static dispensing. In addition, for the two-step method, the amount and composition of the organic cations that can finally be incorporated into the inorganic PbX_6_ (X = I-, Br-, Cl-) octahedral frameworks are determined by the intercalation capabilities of the organic cations. In the first step, PbI_2_ thin films were then obtained by dropping the mixed precursor solution by spin-coating at 3000 rpm for 30 s. Subsequently, the PbI_2_ films were annealed at 120 °C for 5 min in an air environment, and the color of the film turned yellow. In the second step, the mixed FAI–MAI solution was continuously dropped on the pre-coated PbI_2_ film at 8 s intervals during spin-coating at 4000 rpm for 30 s and was subsequently annealed at 150 °C for 10 min. With the increasing spin-coating cycles, only the FAMAPbI_3_ films with 25 mg/mL showed a gradual color change from black to bright red, whereas the films with the 12.5 mg/mL of FAI remained a black color (Figure 1).

To investigate the film morphology, we conducted scanning electron microscopy (SEM, Figure 2a–g). The film morphologies from the lower FAI concentration (Figure 2a–c) varied from 180 to approximately 320 and 370 nm on average for the following coatings. With increasing grain size, the film morphology roughened (Figure 2a–c). For the FA_x_MA_1−x_PbI_3_ films with 25 mg/mL of FAI (Figure 2d–f), the average grain size increased from 160 to 320 nm and 350 nm for successive coatings. In contrast to the similar grain sizes of the films at different concentrations, the morphology was somewhat different. The FA_x_MA_1−x_PbI_3_ film surface with a higher concentration presented wide crevasses between the grains, providing a path for ionic exchange between the FA and MA ions (this will be explained later).

XRD and UV–Vis spectroscopy were performed to investigate the ionic exchanges in the FA_x_MA_1−x_PbI_3_ films. Figure 3a,b shows the XRD patterns of the FA_x_MA_1−x_PbI_3_ films. After the first coating for the FAI of 12.5 and 25 mg/mL, we observed multiple XRD peaks indexed to the (100), (110), (111), (220), and (200) planes of FAMAPbI_3_ [10]. Additionally, we observed the presence of PbI_2_ at a scattering angle of 12.62°, indicating an incomplete conversion of PbI_2_ to FAMAPbI_3_ [11]. After the second coating, the PbI_2_ XRD peak completely disappeared for the 12.5 and 25 mg/mL of FAI. After the third coating, a different structural phase emerged at 12.00°. The peak is responsible for the yellow phase δ-FAPbI_3_, which is consistent with the results of a previous study [11]. To estimate the fraction of FA ratio in the perovskite film, we used the shift of diffraction peak position, and the estimation of the x value was around 0.4. The emerging yellow phase δ-FAPbI_3_ films were also examined using UV–Vis spectroscopy (Figure 3d). For both FAI concentrations, band-edge absorption peaks of FA_x_MA_1−x_PbI_3_ were observed at approximately 760–810 nm (Figure 3c,d) [10]. The bandgap of perovskite is changed indirectly by the mixing ratio of the organic cations because the band structure of perovskite consists of orbitals from Pb and halide [12]. Therefore, we confirmed that the slight change in bandgap of FAMAPbI_3_ of FAI = 12.5 and 25 mg/mL using Tauc-plot (Appendix A). Interestingly, a new absorption peak near 560 nm (2.21 eV) appeared only after the third coating at an FAI of 25 mg/mL (Figure 3d). Because of its energy, δ-FAPbI_3_ is called the yellow phase [13] and is expected to form at FA-rich concentrations. 

In addition, PL spectroscopy was performed to further elucidate the compositional changes of the FAMAPbI_3_ film. For the first coating of FAMAPbI_3_ film with FAI = 12.5 mg/mL, the PL peak appeared at 786 nm, which comes from the cubic α-FAMAPbI_3_ phase [10]. As the number of coating cycles increased, the PL peaks shifted to the longer wavelengths of 790 nm in the fourth cycle of coating. The red shift in wavelengths may be due to the higher incorporation of FA cations into the FAMAPbI_3_ film, resulting in the FA-rich α-FAMAPbI_3_ films (Appendix A). Interestingly, the PL intensity decreased as the number of coating cycles increased. The PL intensity of the first cycle of coating with FAMAPbI_3_ film was the highest among all the samples with FAI = 12.5 mg/mL (Appendix A). It suggests that FA-rich FAMAPbI_3_ is accompanied by defects that increase the non-radiative recombination.

In the case of the FAI of 25 mg/mL, homogeneous spectral peaks and intensities were observed in the FAMAPbI_3_ films with one coating (Figure 4a). No significant differences were observed up to the second coating (not shown). Conversely, in the case of three coatings with the FAI of 25 mg/mL, non-uniform spatial distribution of the PL spectra is shown, as in Figure 4c. In addition, a new PL peak appeared at 589 nm, and the inhomogeneity in the PL mapping increased (Figure 4c). There were two peaks at 589 and 765 nm (Figure 4e). The former corresponded to δ-FAPbI_3_ [14], and the latter was represented by a blue shift of the original peak at 795 nm (Figure 4d). Increasing the FA content induced a new phase in the FA_x_MA_1−x_PbI_3_ films. During dynamic spin-coating, the MA+ and FA+ will possibly compete for incorporation into the PbI_6_ octahedral cages for nucleation [10]. A smaller amount of MA than FA cations is initially advantageous for infiltration. However, FAPbI_3_ has a significantly lower enthalpy value (ΔH_FAPbI3_ = −4.62 eV) than MAPbI_3_ (ΔH_MAPbI3_ = −0.149 eV) [10]. Therefore, with an increase in the number of coating cycles, the surrounding FA+ on the film surface energetically replaces the MA+, forming FA-rich α-FAPbI_3_ perovskite. Subsequently, the energetically favorable δ-FAPbI_3_ overpopulates α-FAPbI_3_ in the films. The formation energy values of δ-FAPbI_3_ and α-FAPbI_3_ are −6.03 and −5.98 eV, respectively [15,16,17].

To estimate the optimum FA cation concentration xFA, which does not introduce δ-FAPbI_3_, we analyzed the variation in the lattice constant of FA_x_MA_1−x_PbI_3_ using XRD by gradually adding FA cations to the films. The estimated distance of the (100) crystal plane, d, of FA_x_MA_1−x_PbI_3_ was 6.330 Å (Figure 3a). By introducing additional FA cations, the distance was modulated and scaled as d(x) = 6.312 + 0.054 × FA (Å) [10,16,17]. Our calculations revealed that the maximum FA cation concentration without δ-FAPbI_3_ was xFA ~0.415.

## 4. Conclusions

We regulated the cation exchange of FA_x_MA_1−x_PbI_3_ films using a dynamic spin-coating method and assessed the effect of the FA concentration on the bandgap modulation. Uniform FA_x_MA_1−x_PbI_3_ films were fabricated using FA concentrations lower than xFA ~0.415, whereas a higher FA composition (xFA > 0.415) resulted in the introduction of δ-FAPbI_3_ into the FA_x_MA_1−x_PbI_3_ films. Our findings attempted to produce perovskite films with more suitable band gaps for Shockley–Queisser limits without the addition of bromide ions. The FA_x_MA_1−x_PbI_3_ film also shows uniform morphology without forming a δ-phase FAPbI_3_, which is of great significance in suggesting a method in which our results can produce an absorber layer to increase the efficiency of the perovskite solar cells.

## Figures and Tables

**Figure 1 materials-14-06422-f001:**
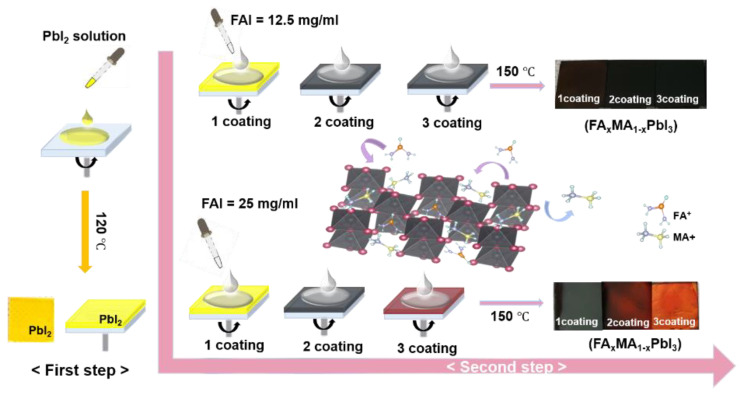
Synthesis of FA_x_MA_1-x_PbI_3_ films using mixed FAI–MAI precursors.

**Figure 2 materials-14-06422-f002:**
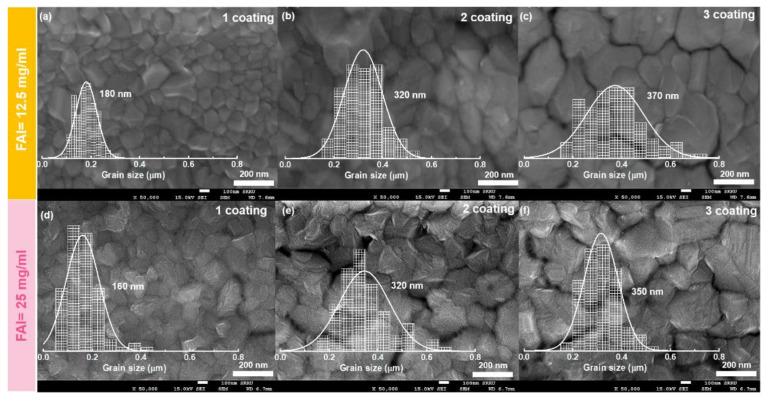
SEM images and grain-size distribution of FA_x_MA_1−x_PbI_3_ films with 12.5 mg/mL FAI after (**a**) 1, (**b**) 2, and (**c**) 3 coatings and with 25 mg/mL FAI after (**d**) 1, (**e**) 2, and (**f**) 3 coatings. The scale bar is 200 nm.

**Figure 3 materials-14-06422-f003:**
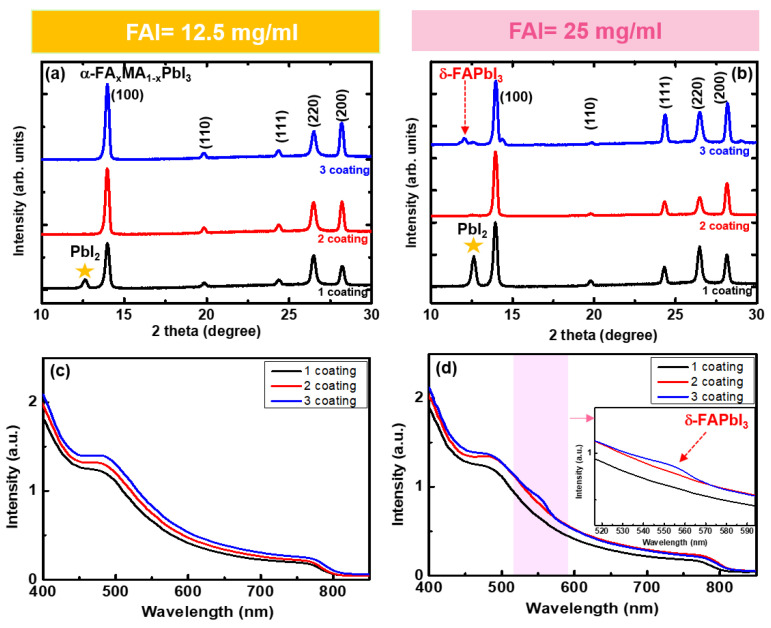
(**a**,**b**) XRD patterns of FA_x_MA_1−x_PbI_3_ from 1 to 3 coatings with FAI = 12.5 and 25 mg/mL, respectively. Absorption spectra with (**c**) 12.5 and (**d**) 25 mg/mL FAI. The inset shows enlarged absorption spectra.

**Figure 4 materials-14-06422-f004:**
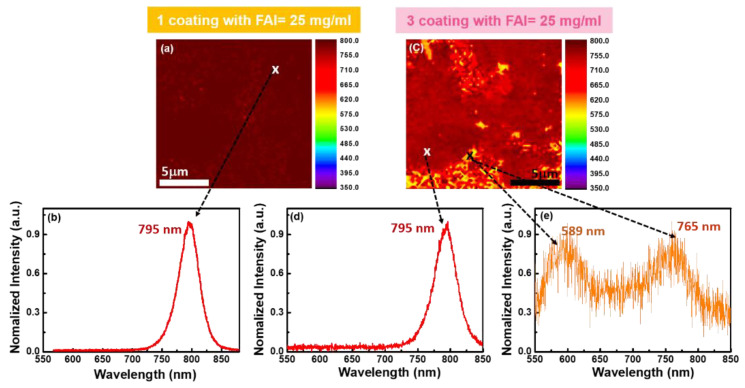
(**a**) PL mapping and (**b**) PL spectra of α-FAMAPbI_3_ with 1 coating of FAI = 25 mg/mL; (**c**) PL mapping of 3 coatings of FAI = 25 mg/mL; corresponding nonuniform PL spectra of (**d**) α-FAMAPbI_3_ and (**e**) showing δ-FAPbI_3_ and FA-rich α-FAMAPbI_3_.

## Data Availability

In this section, please provide details regarding where data supporting reported results can be found, including links to publicly archived datasets analyzed or generated during the study. Please refer to suggested Data Availability Statements in section “MDPI Research Data Policies” at https://www.mdpi.com/ethics (accessed on 25 October 2021). You might choose to exclude this statement if the study did not report any data.

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
