# Peer review of "Investigation of Cation Exchange Behaviors of FAxMA1−xPbI3 Films Using Dynamic Spin-Coating"

_materials, 2021, doi:10.3390/ma14216422_

Round 1

Reviewer 1 Report

The manuscript is interesting and important for application to solar cells. It concerns the research of cation exchange in FAMAPbI3 using the dynamic spin -coating method.

The manuscript can be published following my comments.

My comments:  

 1. Line 85: „”colorless (Figure 1)”  - should be „black color”

  2. In the sentense: „the mixing of MAPbI3 (1.64 eV) and FAPbI3 (1.43 eV) is controlled, the bandgab is tuned to 1.48 eV….[6,7]” :  ref. [6] does not apply to this sentence and there is no reference to the statement that the gap can be tuned to 1.48 eV.

3. Please in Fig.2 to write (a), (b), and (c), (d), (e) but not (a), (b) and (d), (e), (c).

4. Caption Fig. 4 please correct. The caption  (e) is missing. In the the caption ( c) please write  d-FAPbI3. and a-FAMAPbI3.

Author Response

We completed the point-by-point responses to the reviewer’s comments. To improve our manuscript, we revised the manuscript following the reviewer’s comments. We revised the manuscript, figures, and references. Please find below a list of the reviewers’ comments in black, and our response in blue. A revised (resubmitted) manuscript contains changes highlighted in red.

Reviewer1: The manuscript can be published following my comments.

General comments: The manuscript is interesting and important for application to solar cells. It concerns the research of cation exchange in FAMAPbI3 using the dynamic spin -coating method.

Response: We appreciate the reviewer for careful reading of the manuscript and positive comments for the challenging measurements. We have made the following revisions/additions to address the reviewer’s comments. The followings are the details of the response.

Specific comments:

  1. Line 85: „”colorless (Figure 1)” - should be „black color”

Response: We thank the reviewer for pointing out an important aspect. We agree with the reviewer’s comment. Thus, we corrected the word of ‘colorless’ to ‘blak color’ included in line 85 (page2) of the manuscript as follows.

“ With increasing spin-coating cycles, only FAxMA1-xPbI3 films with 25 mg/ml showed a gradual color change from black to bright red, whereas the films with the 12.5 mg/ml FAI remained black color (Figure 1).”

  1. In the sentense: „the mixing of MAPbI3 (1.64 eV) and FAPbI3 (1.43 eV) is controlled, the bandgab is tuned to 1.48 eV….[6,7]” : [6] does not apply to this sentence and there is no reference to the statement that the gap can be tuned to 1.48 eV.

Response: We thank the reviewer for pointing out an insufficient aspect. As the reviewer said, we agree with the reviewer’s comment. We changed the reference No.6, which included that the mixing of MAPbI3 and FAPbI3 is the smaller bandgap than MAPbI3.

6.Luis K. O., Emilio. J. J.-P. and Yabing Q., Progress on Perovskite Materials and Solar Cells with Mixed Cations and Halide Anions. ACS Appl. Mater. Interfaces 2017, 9, 30197−30246.

This reference No.6 exhibited in line 195 to line 196 (page 6) of the manuscript.

  1. Please in Fig.2 to write (a), (b), and (c), (d), (e) but not (a), (b) and (d), (e), (c).

Response: Thanks for reviewer’s comment. As following the reviewer’s comment, we checked the caption of figure 2 and corrected that shown as below:

Figure 2. SEM images and grain size distribution of FAxMA1-xPbI3 films with 12.5 mg/ml FAI after (a) 1, (b) 2, and (c) 3 coatings, and with 25 mg/ml FAI after (d) 1, (e) 2, and (f) 3 coatings. The scale bar is 200 nm.

  1. Caption Fig. 4 please correct. The caption (e) is missing. In the the caption ( c) please write δ-FAPbI3. and α-FAMAPbI3.

Response: We thank the reviewer for pointing out an insufficient aspect. We agree with the reviewer’s comment. Therefore, we corrected the Figure 4. In addition, we wrote the δ-FAPbI3 and α-FAMAPbI3 in the Figure 4 caption as follows.

Figure 4. (a) PL mapping and (b) PL spectra of α-FAMAPbI3 with 1 coating of FAI=25 mg/ml; (c) PL mapping of 3 coatings of FAI =25 mg/ml; corresponding nonuniform PL spectra of (d) α-FAMAPbI3 and (e) showing δ-FAPbI3 and FA rich α-FAMAPbI3.

In addition, the contents of Figure 4 of the manuscript shown on lines 141 to 146 were modified as follows.

Conversely, in case of 3 coatings with FAI of 25 mg/ml, non-uniform spatial distribution of PL spectra is shown in Figure 4(c). In addition, a new PL peak appeared at 589 nm, and the inhomogeneity in the PL mapping increased (Figure 4(c)). There were two peaks at 589 and 765 nm (Figure 4(e)). The former corresponded to δ-FAPbI3 [13], and the latter was represented by a blue shift of the original peak at 795 nm (Figure (d)).

Overall, we thank the reviewer for the comments and valuable suggestions. We have made all necessary revisions based on these comments, which made our manuscript stronger.

Reviewer 2 Report

The work is well edited, the authors have made an effort to answer the questions of referees, indeed the work could be accepted for publication.

Author Response

We completed the point-by-point responses to the reviewer’s comments. To improve our manuscript, we revised the manuscript following the reviewer’s comments. We revised the manuscript, figures, and references. Please find below a list of the reviewers’ comments in black, and our response in blue. A revised (resubmitted) manuscript contains changes highlighted in red.

Reviewer 2

Comments: The work is well edited, the authors have made an effort to answer the questions of referees, indeed the work could be accepted for publication

Response: We thank the reviewer for careful reading the manuscript and positive comments that “Authors have made an effort to answer the questions of referees, indeed the work could be accepted for publication”. We especially appreciate the opportunity to address these questions, and describe the changes we have made accordingly in the manuscript.

Reviewer 3 Report

The manuscript “Investigation of Cation Exchange Behaviors of FA x MA 1-x PbI 3 Films using Dynamic Spin-Coating” describes the fabrication of multi-cation perovskite FA (x )MA (1-x) PbI(3) thin films.

The work is of interest, however, the manuscript must be improved; it looks it was hastily written.

Section “2.1. Material and Sample Preparation” has to be rewritten.

Regarding the section of Results (no. 3) there are several problems.

First question that comes to my mind is - have you tried and other concentations of FAI, others than 12.5 and 25 mg/ml FAI?.

A second question is about the value of x in FA (x )MA (1-x) PbI(3) for the above concentrations.  Do you have an estimation of x?

Analyzing Figures 3c and 3d, there in no significant change in light absorption spectra, just a small shoulder coming from the delta-phase. How do you interpret this? Aparently, there in no bandgap modulation.

Figure 4 is confusing, there is no clear delimitation between 12.5 and 25 mg/ml FAI concentrations.

So, in order to recommend the manuscript for publication the issues raised above must be addressed.

Author Response

We completed the point-by-point responses to the reviewer’s comments. To improve our manuscript, we revised the manuscript following the reviewer’s comments. We revised the manuscript, figures, and references. Please find below a list of the reviewers’ comments in black, and our response in blue. A revised (resubmitted) manuscript contains changes highlighted in red.

Reviewer 3

General comments: The manuscript “Investigation of Cation Exchange Behaviors of FA x MA 1-x PbI 3 Films using Dynamic Spin-Coating” describes the fabrication of multi-cation perovskite FA (x )MA (1-x) PbI(3) thin films. The work is of interest, however, the manuscript must be improved; it looks it was hastily written. So, in order to recommend the manuscript for publication the issues raised above must be addressed.

Specific comments:

  1. Section “2.1. Material and Sample Preparation” has to be rewritten.

We thank for the review comments. we agree with the reviewer’s comment. Therefore, we separated the material section and sample preparation section and rewritten the description of each part in more detail. We give an explanation of the bandgap change in the absorption spectrum in lines 68 to 91 (page 2) of the manuscript as follows.

2.1.1. Preparation of perovksite precusor of FAMAPbI3

First, the mixed perovskite precursor solution was prepared by dissolving 25mg FAI (Greatcell Solar) and 12.5mg FAI (Greatcell Solar) in 1 mL isopropyl alcohol (IPA) solution with 20mg MAI(Greatcell Solar) respectively. And, the 150 mg of PbI2 (lead iodide, 99.9985%) in the mixed solvents of 220 μL dimethylformamide (DMF, Sigma-Aldrich, anhydrous, 99.8%) and 20 μL dimethylsulfoxide (DMSO, Sigma Aldrich) was also prepared. These two solutions prepared in glove box. (glove box condition, O2: 0% and H20: 0%) Afterwards, both solutions were applied on a hot plate at 70 °C for 24 h using magnetic stirring.

2.1.2. Fabrication of FAMAPbI3 fillm.

The FAMAPbI3 perovskite films were fabricated by a two-step dynamic spin-coating procedure on the prepared FTO on glass substrates. FAxMA1-xPbI3 perovskite film was fabricated as follows. The FTO on glass substrate was cleaned sequentially in an ultrasonic bath for 10 min each in acetone, DI water, and IPA, respectively. Afterward, the PbI2 perovskite precursor solution was coated on the FTO/glass substrate by using the dynamic spin coating method at 3000 rpm for 30 s. As a result, the transparent yellow color of the PbI2 thin film was formed. Then, the PbI2 film was heated at 120 °C for 5 min on a hot plate. As a result, the yellow color of PbI2 film was obtained as shown photo image in Figure 1 (a). Finally, the mixed FAI-MAI solution of 50 μL was continuously dropped onto the pre-coated PbI2 films at 8 second intervals during spin-coating at 4000 rpm for 30 s, then annealed at 150 °C for 10 min in an air environment. With increasing spin-coating cycles, only FAxMA1-xPbI3 films with 25 mg/ml showed a gradual color change from black to bright red, whereas the films with the 12.5 mg/ml FAI remained black color (Figure 1 (a)).

Furthermore, we moved the schematic of Fiure 1(a) to Results Part. And we describes the explanation of Figure 1(a) in lines 106 to 119 (page 3) of the manuscript as follows.

Figure1 (a) describes a schematic diagram of the two-step dynamic spin-coating method of FAMAPbI3 film using the mixed FAI-MAI precursor solution. Dynamic spin coating technology was used for continuous dispensing of precursor solutions rather than static dispensing. In addition, for the two-step method, the amount and composition of organic cations that can finally be incorporated into the inorganic PbX6 (X =I -, Br-, Cl-) octahedral frameworks are determined by the intercalation capabilities of the organic cati-ons. In the first step, PbI2 thin films were then obtained by dropping the mixed precursor solution by spin-coating at 3000 rpm for 30 seconds. Subsequently, PbI2 films were an-nealed at 120 °C for 5 minutes in an air environment, and the color of the film turned yel-low. In the second step, the mixed FAI-MAI solution was continuously dropped on the pre-coated PbI2 film at 8 second intervals during spin-coating at 4000 rpm for 30s, which was subsequently annealed at 150°C for 10 min. With increasing spin-coating cycles, only FAMAPbI3 films with 25 mg/ml showed a gradual color change from black to bright red, whereas the films with the 12.5 mg/ml FAI remained black color (Figure 1 (a)).

  1. First question that comes to my mind is - have you tried and other concentations of FAI, others than 12.5 and 25 mg/ml FAI?

Thank you for the review’s comment and appreciate valuable suggestions. Herein, we fabricated FAMAPbI3 film of FAI= 12.5 and 25 mg/ml using dynamic spin-coating with different cycles. Dynamic spin-coating provides more efficient and uniform perovskite films, while there is a lack of understanding of dynamic spin-coating effect on a multi-cation perovskite system. Therefore, we confirmed that FAI = 12.5 and 25 mg/ml of FAMAPbI3 films were coated 1 to 4 times, respectively. Depending on the number of coatings, FAxMA1-xPbI3 films with different FA content ratios are produced. We investigated the cation chage behaviors between FA and MA according to the number of coatings. Furthermore, in this study, an interplay between FA and MA cations was investigated by conducting morphological, structural, and optical analyses.

  1. A second question is about the value of x in FA (x )MA (1-x) PbI(3) for the above concentrations. Do you have an estimation of x?

We thank the reviewer for pointing out an insufficient aspect. For the cation modulation in perovskite film, we used the mixed cation solution in IPA with different mixing ratio of FA and MA. FA is controlled as 12.5 mg/ml and 25.0 mg/ml with MA weight of 20 mg/ml. However, the small size of MA cation has an adavatage of the crystal structure formation, MA cation ratio is more higher in the film than precursor solution. To estimate the cation fraction in the film, we calculated the x value using the shift of diffraction peak and x is approximately 0.4. There is slight change in variation of x depending on the concentration and number of spin coating.

This explantion is included in lines 145 to 146 (page 4) of the manuscript.

To estimate the fraction of FA ratio in perovskite film, we used the shift of diffraction peak position and  the estimation of x value is around 0.4.

  1. Analyzing Figures 3c and 3d, there in no significant change in light absorption spectra, just a small shoulder coming from the delta-phase. How do you interpret this? Aparently, there in no bandgap modulation.

We thank the reviewer for pointing out an insufficient aspect. Generally, the electronic band structure of halide perovskite materials such as MAPbI3 and FAPbI3 are consisted of orbitals from Pb and halides. While, orbitals of organic cations do not participate directly in the perovskite band structure. Therefore, the modulation of band gap is not changed significantly by controlling the ratio of organic cation. However, the size of organic cation affect to the bond angle of Pb-X bond, which modulate the band structure indirectly. In the Figure 3(c) and (d) shows absorption spectra of FAMAPbI3 with different cycle of coating. To demonstrate the small changes of optical bandgap, we investigate the calculation of bandgap using Tauc plot according to the FAI concentration and number of spin coating as shown in Figure SX (added in supplement section). In addition, we give an explanation of the bandgap change in the absorption spectrum in lines 150 to 153 (page 4) of the manuscript as follows.

The bandgap of perovskite is changed by the mixing ratio of organic cations indirectly because the band structure of perovskite is consisted of orbitals from Pb and halide [Nat. Comm., 5, 5757 (2014)]. Therefore, we confirmed that the slight change in bandgap of FAMAPbI3 of FAI= 12.5 and 25 mg/ml using Tauc plot (supplement Figure S1).

Supplement Figure S1. The calculation of bandgap of FAMAPbI3 of FAI=12.5 and 25 mg/ml with different cycle of coating.

  1. Fgure 4 is confusing, there is no clear delimitation between 12.5 and 25 mg/ml FAI concentrations.

We thank the reviewer for pointing out an insufficient aspect. Herein, we confirmed that the formation of delta phase in FAMAPbI3 is only occurred at FAI=25mg/ml. therefore, we conducted the PL mapping of FAMAPbI3 film with FAI= 25 mg. While to avoid confusion between 12.5 and 25 mg, we added the clear explantion of  PL spectra of  FAI= 12.5 and 25 mg/ml. In addition we added the PL spectra of FAMAPbI3 film of FAI=12.5mg/ml in Supplement Figure S2 (a)~(e).  it exhibited in lines 161 to 170 (page 5) and lines 173 to 178 (page5) as follows.

In addition, PL spectroscopy was performed to further elucidate the compositional changes of the FAMAPbI3 film. For 1st coating of FAMAPbI3 film with FAI =12.5 mg/ml, the PL peak appeared at 786 nm, which comes from the cubic α-FAMAPbI3 phase [10]. As a number of coating cycles increased, the PL peaks shifted to the longer wavelengths of 790 nm in 4th cycles of coating. The red shift in wavelengths may be due to the more incorpo-ration of FA cations into the FAMAPbI3 film, resulting in the FA-rich α-FAMAPbI3 films Figure S2 (a). Interestingly, the PL intensity decreased as a number of coating cycles in-creased. The PL intensity of 1st cycle of coating with FAMAPbI3 film was highest among all samples with FAI=12.5 mg/ml (Figure S2 (b)~(e)).) It suggests that FA-rich FAMAPbI3 are accompanied by defects that increase non-radiative recombination. In case of FAI of 25 mg/ml, homogeneous spectral peaks and intensities were ob-served in the FAMAPbI3 films with 1 coating (Figure 4(a)). No significant differences were observed up to the second coating (not shown). Conversely, in case of 3 coatings with FAI of 25 mg/ml, non-uniform spatial distribution of PL spectra is shown in Figure 4(c). In addition, a new PL peak appeared at 589 nm, and the inhomogeneity in the PL mapping in-creased (Figure 4(c)). There were two peaks at 589 and 765 nm (Figure 4(e)). The former corresponded to δ-FAPbI3 [14], and the latter was represented by a blue shift of the original peak at 795 nm (Figure (d)).

Figure S2. PL spectra of FAMAPbI3 film of FAI=12.5mg/ml.                                                          Overall, we thank the reviewer for the comments and valuable suggestions. We have made all necessary revisions based on these comments, which made our manuscript stronger.

Reviewer 4 Report

The authors investigate the “effect of the FA/MA ratio in the crystallinity and photoluminescence properties of FAxMA1-xPbI3 films”, an interesting material for solar cells fabrication. The manuscript is well written, and results are clearly presented. However, the two-step method for preparing the films is already reported in literature. Therefore, I cannot find any point of novelty, especially considering the extensive literature about these films including their performance in devices and novel approaches to enhance the film formation and device performance, see for example:

https://www.sciencedirect.com/science/article/pii/S1876610216315478

https://www.sciencedirect.com/science/article/abs/pii/S1385894721043084

https://link.springer.com/article/10.1007%2Fs40820-020-00418-0

https://onlinelibrary.wiley.com/doi/10.1002/adma.202003801

Therefore, before considering the manuscript, authors should indicate what is the novelty in their work.

Author Response

We completed the point-by-point responses to the reviewer’s comments. To improve our manuscript, we revised the manuscript following the reviewer’s comments. We revised the manuscript, figures, and references. Please find below a list of the reviewers’ comments in black, and our response in blue. A revised (resubmitted) manuscript contains changes highlighted in red.

Reviewer: 4

Comments: The authors investigate the “effect of the FA/MA ratio in the crystallinity and photoluminescence properties of FAxMA1-xPbI3 films”, an interesting material for solar cells fabrication. The manuscript is well written, and results are clearly presented. However, the two-step method for preparing the films is already reported in literature. Therefore, I cannot find any point of novelty, especially considering the extensive literature about these films including their performance in devices and novel approaches to enhance the film formation and device performance, see for example:

[1]https://www.sciencedirect.com/science/article/pii/S1876610216315478

[2]https://www.sciencedirect.com/science/article/abs/pii/S1385894721043084

[3]https://link.springer.com/article/10.1007%2Fs40820-020-00418-0

[4]https://onlinelibrary.wiley.com/doi/10.1002/adma.202003801

Therefore, before considering the manuscript, authors should indicate what is the novelty in their work.

Response:

We thank the reviewer for careful reading the manuscript and pointing out an insufficient aspect. As the reviewer mentioned, there are several reports on preparing the mixed organic cation of organic lead halide perovskite films using either in a one-step (single step) or two-step (sequential steps) spin coating methods. Although the many previous studies of preparation for high quality of multi-cation perovskite film, there are still remaining issues, which are also shown in the references mentioned by the reviewer (example [1-4]). For example, the addition of bromide has been conducted for reducing the spontaneous formation of delta phase (optically inactive phase), even though it leads the bandgap increasing which is not match the Shockley–Queisser limit. On the contrary, in the case of the stabilization of delta phase without bromine ion in perovskite film is achieved, the surface of film is rough which caused the poor interface between perovskite film and charge transport layer locating on the perovskite. The poor interface of light absorber and charge transport layer in the solar cell is the main reason of the degradation of device performance. However, in this study, we demonstrated the stabilization of alpha phase of FAMAPbI3 perovskite film without d-phase FAPbI3. At the same time, we also achevied a smooth surface of the α-phase FAMAPbI3 perovskite film without bromine ion treatment. Herein, we fabricated multi-cation FAMAPbI3 film by controlled FA/MA compositions using two-step dynamic spin-coating, which avoids fast crystallization and gives better control over the conversion process of the perovskite layer, resulting in high reproducible film morphologies. In addition, by conducting the  morphological, structural, and optical analyses, it is found that dynamic-coating cycle and FAI composition influenced film quality and developed the d-phase of FAPbI3 within FAMAPbI3 film. We believe that the result of this study has a novelty to guide the way for accomplishing the high performance of perovskite solar cells.

Therefore we corrected the conclusion parts as follow. This explanation is included in lines 199 to 207 (page6)

We regulated the cation exchange of FAxMA1-xPbI3 films using a dynamic spin-coating method and assessed the effect of the FA concentration on the bandgap modulation. Uniform FAxMA1-xPbI3 films were fabricated using FA concentrations lower than xFA ~0.415, whereas a higher FA composition (xFA > 0.415) resulted in the introduc-tion of δ-FAPbI3 into the FAxMA1-xPbI3 films. Our findings attempted to produce perovskite films with more suitable band gaps for Shockley–Queisser limits without the addition of bromide ions. The FAxMA1-xPbI3 film also shows uniform morphology without forming a δ-phase FAPbI3, which is of great significance in suggesting a method in which our results can produce an absorber layer to increase the efficiency of perovskite solar cells.

Overall, we thank the reviewer for the comments and valuable suggestions. We have made all necessary revisions based on these comments, which made our manuscript stronger.

Round 2

Reviewer 4 Report

Authors have addressed the comments from reviewers.

This manuscript is a resubmission of an earlier submission. The following is a list of the peer review reports and author responses from that submission.

Round 1

Reviewer 1 Report

Authors use dynamic spin-coating to fabricate mixed cation perovskite films. Similar methods have been used by many researchers and reported in literature; an organic layer is fabricated on a metal halide layer, and, then, the two-layer sample is heated to covert to the perovskite. The reviewer does not think the authors’ method is original and special to improve the perovskite film quality. Additionally, the authors do not cite the previously reported papers describing similar methods. The reviewer and readers want to see photovoltaic performance of the authors’ perovskite films but such results are nothing. The reviewer cannot suggest the present manuscript for publication in the IF=5 journal.

Reviewer 2 Report

In this paper, authors reported the Investigation of Cation Exchange Behaviors of FAMAPbI3 Films using Dynamic Spin-Coating. This work is interesting and short, but parts of the concept are not novel. However, it needs major revision before it can be accepted.  Some of the corrections and suggestions are as follows:

  1. The writing in Introduction part is not good. I recommend rewriting and exhibit your innovation.
  2. The scale bar of the SEM is not clear readers.
  3. In the XRD section, some peaks are not indexed. Please index these character peaks.
  4. The authors did not describe the choice of substrate and did not talk about the method to measure thickness of the layers.
  5. Correct any typo such as line 75 (Cu-kα),
  6. In Fig.1, ry to differentiate between samples of FAMAPbI3, the image is unclear.
  7. Please in Fig.2, try to write (a), (b), (c) and (d), (e), (f) but not (e), (f), (g)
  8. Please legends are not clear in Fig. 4.
  9. There are some key and important perovskite research findings that need to be mentioned and cited so that we can provide a solid foundation and progress for readers, such as DOI: 1007/s10854-020-05078-9.
  10. Why the author uses the word dynamic spin coating and not spin coating in this work?
  11. Try to keep the same handwriting in the manuscript (4000 rpm or 4,000 rpm),
  12. There is no word nanomaterial in the manuscript, yet the journal title is nanomaterial, do the authors have an explanation?

Reviewer 3 Report

The manuscript is interesting and important for application to solar cells, however, I have a few comments

  1. A description of the experiment is included in section 3 “Results and Discussion”. This part should not contain a description of the experiment, moreover, this description is provided in part 2.1 “Material and sample preparation”.
  2. Please add a reference to the sentence that bang gap of MAPbI3 is equal 1.64 eV. According to my information Eg = 1.55-1.613 eV.
  3. Please explain effect of gaps modulation in the conclusions. The maximum value of PL is 795 nm, which corresponds to the energy gap of 1.55 eV, which is the same as for MAPbI3. So, what is the profit on widening the energy gap?
  4.  Please check the reference numbering carefully. Instead of [6], it should be [7].
  5. The numbering in Figure 2: (e) is missing.
  6. I propose to write FAxMA1-x PbI3 instead of FAMAPbI3.
  7.  Line 18: homogeneity of the photoluminescence spectra, should be homogeneity of the photoluminescence spatial distribution.
  8. Figure 4 – the indications (a) and (b) are not visible